# WaveletGPT: Wavelets Meet Large Language Models

## Abstract

Large Language Models (LLMs) have ushered in a new wave of artificial intelligence advancements impacting every scientific field and discipline. They are trained on a simple objective: to predict the next token given the previous context. We live in a world where most of the data around us, e.g., text, audio, and music, has a multi-scale structure associated with it. This paper infuses LLMs with traditional signal processing ideas, namely wavelets, during pre-training to take advantage of the structure. Without adding **any extra parameters** to a GPT-style LLM architecture in academic setup, we achieve the same pre-training performance almost twice as fast in text, raw audio, and symbolic music. This is achieved by imposing a structure on intermediate embeddings. When trained for the same number of training steps, we achieve significant gains in performance, which is comparable to pre-training a larger neural architecture. Our architecture allows every next token prediction access to intermediate embeddings at different temporal resolutions in every Transformer decoder block. This work will hopefully pave the way for incorporating multi-rate signal processing ideas into traditional LLM pre-training. Further, we showcase pushing model performance by improving internal structure instead of just going after scale. [1]

## 1 Introduction and Related Work

Large Language Models (LLMs) have ushered in a super-renaissance of AI models and are touching every scientific and engineering discipline. At the heart of this revolution has been the Transformer architecture (Vaswani et al., 2017), initially proposed for machine translation in natural language processing. Transformer architecture became the backbone of GPT (Generative Pretrained Transformer) language models (Brown et, 2020) first proposed by Open-AI, which has revolutionized the field. Modern LLMs are still trained on a straightforward objective: To predict the next token given the previous context, preserving the causality assumption. The exact recipe has been shown to work not only for language but also for robotics (Brohan et al., 2023b;a), protein sequences (Madani et al., 2020), raw audio waveforms(Verma & Chafe, 2021), acoustic and music tokens (Huang et al., 2019; Verma & Smith, 2020; Borsos et al., 2023), videos (Yan et al., 2021) to name a few. This simple recipe of tokenization/creating an embedding and feeding it to transformers also has given rise to architectures in non-causal setups such as BERT(Devlin et al., 2019), Vision Transformers (Dosovitskiy et al., 2021), Audio Transformers (Verma & Berger, 2021) and Video Transformers (Selva et al., 2023). The recent surge in multi-modal large language models similar to that proposed by Google with its Gemini family (Team et al., 2023) or multi-modal models like Chameleon (2024) would pave the way for another wave of applications in the future. With increased performance by scale, some of the models like GPT-3 are reaching hundreds of billions of parameters (Brown et, 2020) to that of Google's Switch Transformer has even reached trillion parameters (Fedus et al., 2022). This has led to recent concerns that AI research is slowly moving out of academics and is getting confined to industry researchers, as per the recent Washington Post article written by Nix (2024).

The theme for this work is to push the capabilities of the models to get capabilities of a much bigger architecture or achieve the same performance in smaller training steps. Briefly, we take intermediate embeddings after each of the decoder block, and without adding any parameters, impose a multi-scale structure that represents a hierarchy. We find signals in the intermediate embeddings across tokens. To each of these signals as

---

[1]* This work was carried out while XXXX was with the XXXX XXXX in the XXXX XXXX XXXX at XXXX XXXX

explained in Figure 1, we similar to a traditional wavelet decomposition, modify it resembling an approximate version of the signal retaining causality assumption. Researchers have proposed several techniques to boost the performance of smaller architectures using larger models. Our work differs from some of the techniques discussed next, and we propose improving performance during pre-training. One of the most popular ones is knowledge distillation Hinton et al. (2015), where a larger model in terms of the number of parameters is used to guide a smaller architecture. Gu et al. (2024) used KL divergence-based criteria to improve the capabilities of generating text (next token prediction) from the teacher model's feedback. This still uses a powerful model to improve its performance rather than improve the smaller architecture trained from scratch. A line of work also proposed hierarchical transformers via upsampling-downsampling operation proposed by Nawrot et al. (2022) similar to an hour-glass U-Net architecture in computer vision by Long et al. (2015). As compared with the Transformer baseline, given the same amount of computation, it can yield the same results as Transformers more efficiently. Our work has similarities and stark differences to Clockwork-RNN (Koutnik et al., 2014). First proposed for improving long context modeling in RNNs, it splits the hidden neurons of an RNN into different modules, and each module, having different parameters, updates their states at different (clock) rates. Thus, at each time step, only a few modules(weights) are activated and updated during forward/backward passes. This allows the network to learn dependencies through processing and retaining long-term information at different rates from the high-speed and low-speed modules. Our architecture only tinkers with the intermediate embeddings with straightforward tweaks and does not introduce complex separate learning modules or update weights at different rates. Model pruning (Sun et al., 2024), on the other hand, removes weights based upon its saliency in impacting the performance to achieve the same performance of the large architecture, like LLAMA (Touvron et al., 2023) with fewer compute flops, during inference. Again, the goal is to start with a trained large model at the outset rather than trying to achieve the same pre-training performance from scratch. We also do not discuss quantization-based algorithms proposed by Dettmers et al. (2024), as they are also focused on improving inference times/flops or fine-tuning a pre-existing architecture.

Like ours, the other line of work is tinkering with the intermediate embeddings. Tamkin et. al (2020) proposed hand-tuned filters on the Discrete Cosine Transform (Ahmed et al., 1974) of the latent space for different tasks like named entity recognition and topic modeling for non-causal architectures like BERT (Devlin et al., 2019). However, they take a discrete cosine transform over the entire context length. They thus cannot be adapted for applications such as language modeling, which predicts the next token in a given context. There have been similar papers on applying ideas from signal processing-like methods to BERT-like non-causal architectures, and we will discuss two of them here, FNet and WavSPA, that are relevant to our current paper, both again proposed for BERT-like architectures. Both papers present variants of improving attention blocks, which is different from our work on causal decoder-only architectures such as GPT. FNet proposed by Lee-Thorp et al. (2022) removes the costly attention mechanism and replaces it with a 2-D FFT block. This operation is, however, non-causal as it looks into future tokens for computing 2-D FFT for modifying the current tokens. On the other hand, WavSpA (Zhuang et al., 2024) computes the attention block in the wavelet space. The premise is that since wavelet transform is a multi-resolution transform capturing long-term dependencies at multiple time scales, the input sequences are transformed into wavelet space, and the attention mechanism is carried out and then reconstructed. However, one of the significant drawbacks is that the operation is non-causal, i.e., to compute the wavelet transform, one needs to look at the entire sequence length for capture variations from coarsest to finest scales (as can be seen in Figure 1 of (Zhuang et al., 2024)). Thus, such modifications cannot be adapted to GPT-like decoder-only architectures. As we will see in our work, we modify only the intermediate embeddings, leaving the rest of the architecture the same as is in a causal manner. Our work is also inspired by neuroscience, which provides evidence that the human brain learns multi-scale representations for language at multiple time scales (Caucheteux et al., 2023) instead of fixed resolution representations. As we will see in our work, our paper explicitly proposes imposing multi-scale representation during pre-training onto every intermediate decoder embedding.

The contribution of the paper is as follows:

- We propose, to the best of our knowledge, the first instance of incorporating wavelets into large language models. We propose the addition of multi-scale filters onto each of the intermediate embeddings of Transformer decoder layers using Haar wavelet. Our architecture allows every next

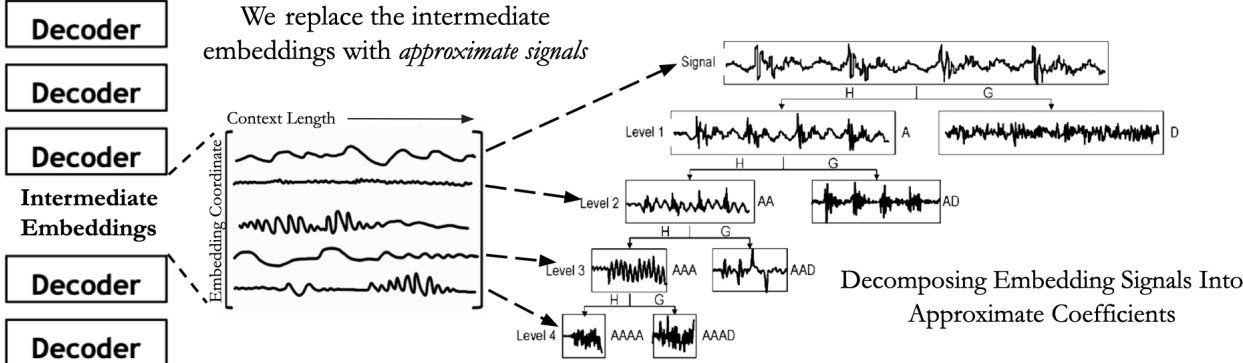

Figure 1: We manipulate signals in every intermediate embedding between decoder blocks of GPT. For each of these signals of length equal to the context length of GPT, we compute a simple 1-D discrete haar wavelet transform at different levels to approximate the signal at different resolutions to mimic the multi-scale structure that exists in the real world for text, raw audio, and symbolic music. The figure on the right is from Gao & Yan (2006), which gives a more detailed account of non-stationary signal processing for 1-D signals. We go on the leftmost route of approximate coefficients, which allows us to capture embeddings at different resolutions.

token prediction access to multi-scale representations for the intermediate embeddings in every Transformer decoder layer instead of fixed-resolution representations.

- We show that adding no extra parameter can substantially speed the pre-training of a transformer-based LLM in the range of 40-60%. This finding is substantial given how ubiquitous the Transformer Decoder-based architectures are across various modalities. We also show that with the same number of training steps, the model gives a substantial non-trivial performance boost, akin to adding several layers or parameters.

- We show that adding a wavelet-based operation gives a performance boost in three different modalities for the pre-training tasks regarding validation loss. These three modalities are language (text-8 (Mikolov et al., 2012)), raw audio (YoutubeMix (Goel et al., 2022)), and symbolic music (MAESTRO (Hawthorne et al., 2019)). This shows that our method is generic enough for structured datasets.

- We also explore that by making these kernels learnable, which adds only a tiny fraction of the parameters, as compared to the primary model, we get an even further increase in the performance of our model, which allows the model to learn multi-scale filters on the intermediate embeddings from scratch.

## 2 Dataset

We utilize three open-source datasets to showcase the strength of our proposed method. In addition, we choose them from three different domains: natural language, symbolic music, and raw audio waveform. For text, we choose text-8 (Mikolov et al., 2012). We choose this over other datasets as i)it is a popular and widely cited character-level language modeling dataset for text and ii) in order to use a simple vocabulary (space + 26 lowercase characters) to detach the effects of various tokenizers from our results, at least in one of the modalities. It has 100M characters with the split used in training, validation, and testing as given by Al-Rfou et al. (2019). We report the results for two modalities other than text: raw waveform and symbolic music. For raw audio, the goal is again to predict the following sample given a context of samples. We use the YouTube-Mix-8 dataset, which has been used for long-context modeling (Goel et al.,

2022; Verma, 2022). Here, since we use 8-bit signals, our vocabulary size is 256, with a sampling rate 16KHz. We use a third dataset, MAESTRO (Hawthorne et al., 2019), which contains over 1000 MIDI files of popular classical music pieces. We use Google's tokenizer proposed by Huang et al. (2019), which can convert MIDI tracks into discrete tokens with a vocabulary size 388. An important point to note is that the goal in all three modalities is not to chase state-of-the-art performance, as *this paper was written in an academic setting with very few computational resources at disposal* . The goal is to shrink GPT-like architecture and compare pre-training performance to the shrunk-down version with/without adding multi-scale structure to the embeddings,**without adding any extra learnable parameter.**

## 3 Methodology

This section will describe the approach to incorporating wavelets into transformer-based Large Language models while retaining the causality assumption. The ideas described here are generic and can be easily extrapolated to setups without a Transformer architecture.

### 3.1 Incorporating Wavelets into Intermediate Embeddings

For any signal, we would compute one of the versions of discrete wavelet transform as we will describe and incorporate that back into the signal. Let us assume that $x^l_{(i)}$ is the output of the $l^{th}$ decoder layer and represents the activation along the $i^{th}$ coordinate. This activation signal will have a dimension equal to the context length of the transformer-based GPT model. In our case, we denote the context length as $L$. So now, if in the original GPT architecture, there were $N + 1$ layers, with the embedding dimension as $E$, we would get $N.E$ signals of length $L$ from all of the intermediate embeddings between two decoder blocks. $E$ in our case goes from $[0 - 128)$ dimensions.

### 3.2 Introduction to Wavelets

A wavelet is a signal that typically has zero mean and a non-zero norm. A wavelet transform was first designed to overcome the shortcomings of a traditional Fourier-based representation. Given any signal $x[n]$, a discrete wavelet transform is akin to passing the signal through filters with different resolutions, as shown in Figure 2. In its simplest form, we will use Haar wavelet, a family of square-shaped functions throughout this paper. The family is obtained from a mother wavelet via scaling and shift operations. Given a mother wavelet function $\psi$, we come up with the child wavelets as

$$\psi_{j,k}[n] = \frac{1}{\sqrt{2^j}} \psi \left( \frac{n - k2^j}{2^j} \right) \tag{1}$$

Here $j$ is the scaling factor and $k$ the shift factor. These are nothing but signals that are shifted, and scaled to capture the information of the signal of interest at various time-scales, with $n$ being time or in our case the context length. This should immediate strike similarity to that of the diagram in Figure 1 to capture various signals present in Transformer decoders intermediate layers at various resolutions. We now define discrete wavelet transform. Simply, it can pass any signal through filters and downsampling operations. This operation, as seen in Figure 2, should immediately strike a resemblance to a convolutional neural net like Resnet (He et. al, 2016), which consists of learned convolutional filters analogous to $h[n]$ and $g[n]$, and downsampling operation like max-pooling. In traditional state-of-the-art convolutional architecture, we typically follow one branch of Figure 2, i.e., we take the output of filters, downsample, and do it recursively. This was also one of the reasons wavelets were incredibly popular in the early 90s and 2000s for image understanding, as one can see parallels to that of convolutional architectures (Huang & Aviyente, 2008; Kingsbury & Magarey, 1998). Let us assume that we choose a family of wavelets (Haar wavelet in our case); then, it would be akin to passing the signal through a low-pass and a high-pass filter corresponding to the kernels in that family of wavelet transforms $g[n]$ and $h[n]$ respectively. In the case of Haar wavelet transform, it is simply taking averaging and difference operation, i.e., the impulse response of $g[n]$ and $h[n]$ are [1/2,1/2] and [1/2,-1/2] respectively. Let us look at Figure 2 for a more detailed explanation of a discrete wavelet transform. Let $x[n]$ be any 1-D length signal $L$. In order to get level 1 coefficients, we pass it through two

filters with impulse response $g[n]$ and $h[n]$ followed by a downsampling operation. Thus, the approximation coefficients $y_{approx}$ and $y_{detail}$ are simply the output of an LTI system defined by a convolution followed by downsampling (by two here) defined as in Equation 2. This is the reason, we in Algorithm 1 have convolution operation kernel mimicking this behaviour.

$$y_{\text{approx}}[n] = \sum_{k=-\infty}^{\infty} x[k]g[2n-k] \quad ; \quad y_{\text{detail}}[n] = \sum_{k=-\infty}^{\infty} x[k]h[2n-k] \tag{2}$$

Now, in order to get multi-scale representations of the original signal, the same operation that is described above for $x[n]$ is carried out recursively now for $y_a$ (approx) to get level 2 wavelet coefficients $y_a^2$ and $y_d^2$ (detail) and so on. In our case $x[n]$ are the intermediate signals across the context length in each of the coordinates at the output of every decoder block present in the LLM. Typically, the collection of signals describing approximate coefficients $y_a$ and $y_d$ along with their decomposition namely, $y_a, y_d, y_a^2, y_a^3, y_a^4...$ are used for further processing for various application. We note that $y_a^2, y_a^3, y_a^4$ will have smaller lengths by a factor of 2, 4, 8, and so on. For the Haar wavelet transform, we can recursively go down the route of approximate coefficients and average the adjacent two samples. We can take the average of the current and past samples to retain the causality assumption, as we will see soon. We can see that higher-order approximate coefficients capture averages at much larger context length if we keep going the route of only the approximate coefficients, as seen in Figure 2. We can go to maximum depth till we are only left with a single value, a scalar, that is representative of the entire signal, which is the mean over the length of the signal. Haar wavelet transform computes averages and differences of the signal to get a multi-resolution representation of the signal, capturing low and high frequencies of the signal at different resolutions. This can be seen in Figure 2, where the same signal on the right is captured at the coarsest representation and then finer detail representations using Haar wavelets. We do this on the intermediate embeddings, allowing every next token prediction access to such representations.

### 3.3 Connecting wavelets and LLM embeddings

Often, in many signal processing applications, the first-order detail coefficients and all of the approximate coefficients are used to understand the contents of the signals at various levels. We also intend to carry out the same operation, but we are now getting signals from intermediate transformer embeddings. However, we do not take detailed coefficients and look into the approximate ones. This was our premise: that real-world data around us is structured. For text, the structure at different levels ranges from letters, words, sentences, paragraphs, topic models, etc. In the case of symbolic music, it can be thought of as musical notes to motifs to pieces and so on. Since we chose Haar wavelet for the rest of this work, this can be approximated as a simple averaging operation, as described in the previous section. If we keep going down the path of the approximate coefficients, we will eventually have only a single scalar, which is the average of the whole signal for the case of the Haar wavelet. In order to get the same sequence length from the approximation coefficients as the original signal, there can be several ways, with up-sampling the signal back to the original length being one of them. As part of nomenclature, we call the signal approximated at a particular level with the same length as the *"approximate signal"* at that level to discern it from the approximate coefficients, which *are smaller in length.* In Figure 2 (R), in order to get the signal approximation at various levels, which is equal to the original input signal $x[n]$, the wavelet kernel being averaging operation, we take the approximate coefficients and multiply it with the kernel at that level. ([1,1], [1,1,1,1], .... and so on). This can be reflected in the piece-wise constant function, as seen in Figure 2. We can see that for different embedding coordinates of LLM embeddings, we define different resolution kernels, each of them corresponding to a particular scale at which we should capture the data. The reconstructed signal $x_{recon}[n]$, which is one way of getting the *approximate signal*, is computed from its wavelet coefficients at a particular level $j$ as $c_j$ as,

$$x_{\text{recon}}^j[n] = \sum_k c_k \cdot \psi_{j,k}[n] \tag{3}$$

Notice that Equation 3 requires storing the child wavelets are various approximations and using them with the approximate coefficients to get the signal back. However this is a complex operation and is non-causal. In order to adapt it to LLM and Transformer decoder architecture, we simplify computing the *approximate signal* in a differentiable manner. We opt for a simple variant of the equation describe in Equation 3.

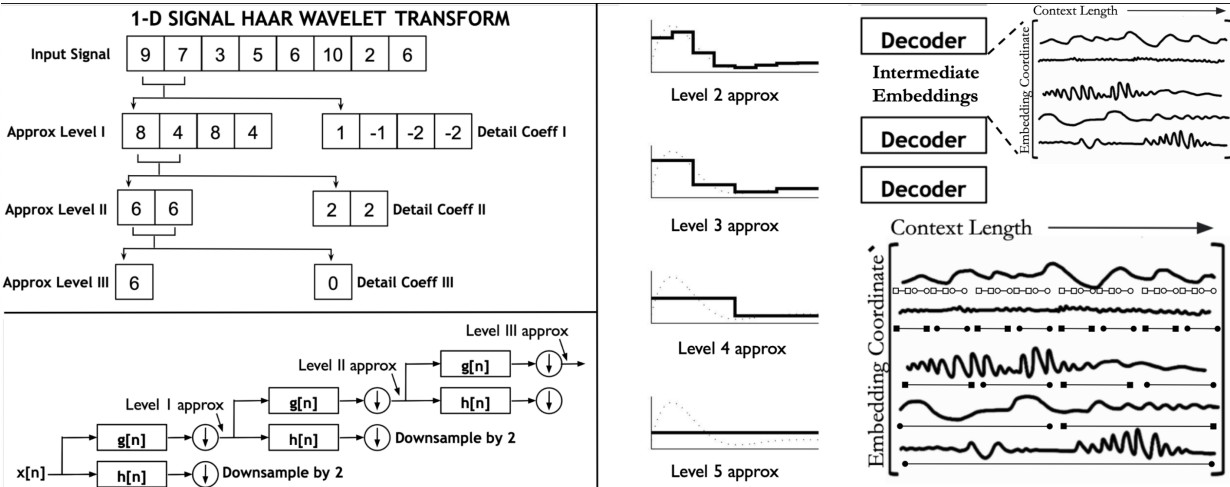

Figure 2: (Bottom left): A tree structure depicting a 3-level filter bank that can take the signal and give us different resolutions of the signal. In this case, we only focus on the approximate coefficients by passing it through an impulse response corresponding to the wavelet of choice and recursively down-sampling it. (Top left) How to compute the approximate and the detailed coefficients at various levels through a toy example. We recursively take first-order averages/differences followed by downsampling until we get only a single scalar representative of the input signal. (Right) For a 32-length signal, we depict how different levels of approximate coefficients of discrete haar wavelet transform to capture the signal from the coarsest to fine details. The figure on (center) and (top-left) are redrawn from a tutorial on wavelet transform from (Flores-Mangas, 2014). (right) How our architecture computes embeddings moving at different rates via the wavelet approximation where for some of the embedding dimensions, the information moves at the coarsest rate similar to level 5 approximation, whereas, for other dimensions, it follows finer resolution similar to a level 2 approximation. Notice during next token prediction this multi-scale knowledge is present in all decoder layers.

In the case of the Haar wavelet, a simple averaging operation, we take the moving average of the input signal with varying kernel length. We keep increasing the kernel's length for averaging until it becomes that of the context length (when a single scalar approximates the whole signal). The kernel length decides which signal approximation level we are interested in. Since LLMs operate on causality assumption for any input signal and a given kernel length, if needed, we get the modified value of the signal at a location by computing the moving average of the prior samples of the input signal within the kernel length. We zero-pad the signal to the left to account for the cases and token dimensions when the length of the signal is less than that of the kernel. The discrete Haar wavelet transform at different levels gives multiple versions of the same signals. This might create more copies of the same signal and mess up the structure and the dimensions of the intermediate Transformer embeddings. In order to avoid this issue, we create different resolutions for different approximations of the signals. In Section 4.4, we make these kernels learnable, thereby allowing the architecture to retain the same operations as moving average, but instead of the kernel being a constant, we allow it to be learnable. The resolution at which we look at the signal is now parameterized by the coordinate in the model dimension, which will be explained in more detail in the next section.

### 3.4 Wavelet Coefficients by Embedding Dimension Coordinates

One option would be to take each of the signals $x^l_{(i)}$ in each of the coordinates of every decoder layer and compute each of their *approximate signals* in level $I$, $II$, $III$, $IV$ and so on. This would have exploded the number of signals that we have. For example, for a context length of 512, we would need nine more copies with a resolution of 512, 256, 128, 64, 32, 16, 8, 4, and 2 describing level $I$ to $IX$ coefficients of the original signal. This would tremendously increase the complexity of our architecture, in our case, a GPT, and would have required significant architectural changes to incorporate an increase in information via multiple

---

**Algorithm 1** Wavelet-GPT

---

$E$: Model or Embedding Dimension
$L$: Context Length
$N + 1$: Number of Decoder Layers
**for** layer $l = 1, 2, \ldots, N$ **do**
    $\mathbf{x}^l \leftarrow$ Output of Transformer $l^{th}$ Decoder Block                               //Dimension $E$ x $L$
    $\mathbf{xn}^l \leftarrow$ Modified Transformer Embedding Replacing $\mathbf{x}^l$
    $\mathbf{xn}^l_{(i)} \leftarrow \mathbf{x}^l_{(i)}$    For Embedding dimension    $i > E/2$

    $\mathbf{f(i)} \leftarrow 2^F$ where     //Finding kernel length a function of embedding coordinate as nearest power of 2
          $F = int(L_k * (i - E/2)/(E/2 - 1))$    $L_k = \lfloor \log_2(L) \rfloor + 1$    $i <= E/2$

    $\mathbf{xn}^l_{(\mathbf{i})}(\mathbf{k}) \leftarrow \frac{\mathbf{1}}{\mathbf{f(i)}} \sum_{\mathbf{m=k-f(i)}}^{\mathbf{k}} \mathbf{x}^l_{(\mathbf{i})}(\mathbf{m})$    $i <= E/2$          //For Non-learnable fixed Haar wavelet
    $\mathbf{xn}^l_{(\mathbf{i})}(\mathbf{k}) \leftarrow \sum_{\mathbf{m=0}}^{\mathbf{f(i)-1}} \mathbf{h(m)} \cdot \mathbf{x}^l_{(\mathbf{i})}(\mathbf{k-m})$    $i <= E/2$    //For learnable multi-resolution wavelet kernel $h$
**end for**

---

additional resolution signals. To mitigate this, we come up with a novel solution as follows: We do not compute all levels of *approximate signal* for each intermediate embedding dimension signal across tokens. We parameterize the level to be computed for the *approximate signal* by the index of the embedding dimension itself. Another important thing is that we want to steer the embeddings only a little into the inductive biases we impose. Transformers have been wildly successful without incorporating any inductive biases. Ideally, we want the best of both worlds, nudging intermediate GPT embeddings in only half of the dimensions. For this, we retain half of the intermediate embedding signals along the coordinate dimension at the exact resolution, i.e., with no change. For the embedding coordinates from 64 to 128 ($E/2$ to $E$), when the model dimension is 128, we do not do any processing or manipulations. For the other half, we do some processing parameterized by their index $i$. Mathematically, if $x^l_{(i)}$ is an intermediate embedding after the $l^{th}$ decoder layer along the $i^{th}$ coordinate dimension, then for half of the coordinate dimensions of the modified new signal $xn^l_{(i)}$ will remain same as that of $x^l_{(i)}$ for $i$ from $E/2$ to $E$. For the second half, we impose our structure by using **approximate signal** at a particular level. This is mainly because Transformers are quite expressive, and we want to avoid too much tinkering with what they learn. For the other half, $xn^l_{(i)}$ is the modified latent space that is obtained from $x^l_{(i)}$ by first getting the wavelet coefficient level corresponding to that of the coordinate $i$. We use a simple mapping function $f$ to take the coordinate dimension $i$ as an argument. In our case, $f$ takes in the argument from $i$, ranging from 0 to $E/2$ (0-64), and returns the kernel size corresponding to the approximation coefficient level between $I$ and $IX$. We use a simple linear function that slowly increases the index from $I$ to $IX$ between 0 to $E/2$. So when $i$ is 0, $f(i)$ maps to level $I$ approximation kernel 2, and when $i$ is $E/2$ (64 in our case), $f(i)$ maps to level $IX$ approximation kernel of length 512, (or for generic case, the level that would be corresponding to the coarsest representation, i.e., a single scalar). Now, let us find out how we compute the modified new signal $xn^l_{(i)}$ that replaces the original intermediate Transformer embeddings $x^l_{(i)}$. $f(i)$ denotes the kernel size for the coordinate $i$. Now, the modified signal is as follows:

$$ xn^l_{(i)} = x^l_{(i)} \text{ for } i > E/2 \quad ; \quad xn^l_{(i)}(k) = \frac{1}{f(i)} \sum_{m=k-f(i)}^{k} x^l_{(i)}(m). \tag{4} $$

For the cases where $k - f(i) < 0$, we zero-pad the signal to make this signal valid for the average to be computed. In other simple words, for the case of the Haar wavelet, the modified signal is nothing but a causal moving average filter that computes the average of values of the embedding signal along $i^{th}$ coordinate with a kernel size as a function $f(i)$. As described in the Equation 4, this simple algebraic operation does not add any parameters to the architecture. This retains the causality assumption critical in LLM and prevents any leakage from happening to the future tokens from any of the embedding dimensions. We can adapt this so that instead of simple averages akin to Haar wavelet as described above in Equation 4, we allow it to learn an optimal kernel specific to the problem at hand. We explain this in Algorithm 1, where each value of the modified signal at a token $k$, is a convolution with a learned kernel $h(.)$, with variable kernel length $f(i)$,

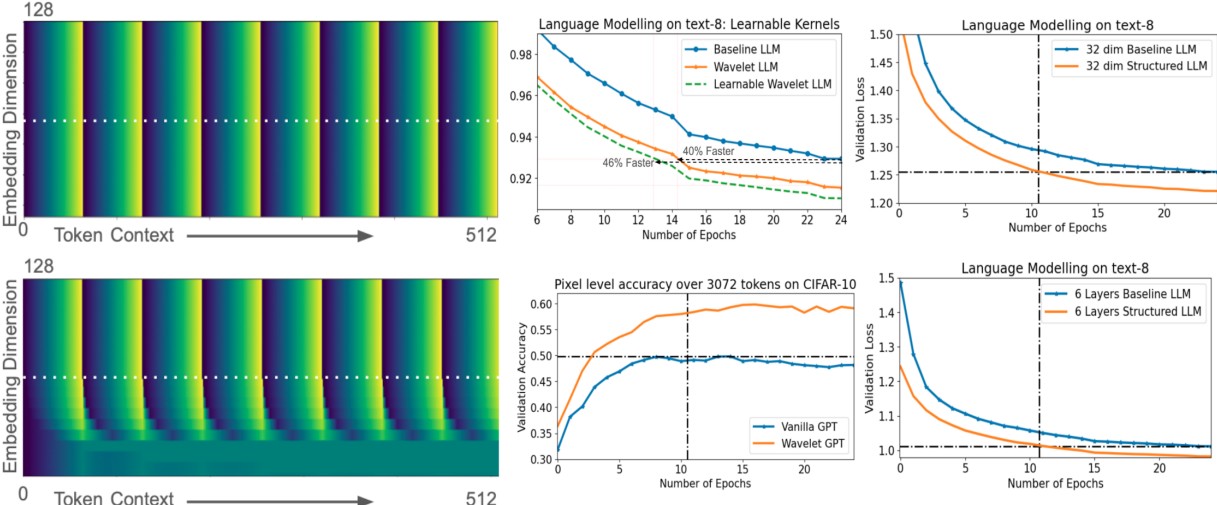

Figure 3: (Left) A toy example where variations of embeddings shown via heat-map move along a token dimension and how our processing imposes a multi-rate structure where different embedding dimensions move at different rates while retaining the causality assumption. This allows intermediate latent space to learn information moving at different rates at every token. Notice how the pattern disperses from dimension 64 to 0. (Right) Validation loss of LLM is pre-training on text-8 with the addition of the structure, making the kernels learnable. We can see that we achieve the same performance almost twice as fast. When trained for the same number of epochs, we get a performance boost akin to adding additional decoder layers. We also show how our architecture behaves on text-8 for the smaller model dimension of 32, retaining the performance speedup as a 128-dim model and shallow depth version with six layers and 128 model dimensions. We also see that for the Long Range Arena image benchmark, we get a boost of 10% with the addition of no extra parameters.

which is parameterized by the coordinate dimension $i$ of the embedding dimension. We learn these kernels independently for every signal present in the LLM.

## 3.5 Imposing Structure: Toy Example

As we can see from Figure 3 we have shown a toy example to depict how we impose a structure onto decoder Transformer embeddings. In Figure 3 (left), on top, eight variations along the token dimension are present, with onset (largest values or sudden bursts) at token index numbers 32, 64, and so on and decreasing to zero in the next token and then again increasing to the largest value linearly till the next interval. As motivated in the introduction before, datasets around us have an inherent structure present in them. In order to capture this structure, we impose a structure onto intermediate Transformer embeddings in every layer. For the toy example, we can see from Figure 3 (left), no bottom, that we retain the embeddings at the exact resolution for half of the embedding dimensions (split by white line). For the other half of the embedding dimension, we slowly increase the kernel length across the context length and causally compute the average. We reach the last embedding dimension, which moves the slowest and takes the average across the token dimension with the kernel size equal to the context length (zero-padding the signal if necessary). This creates highways that allow some coordinates of the embeddings to move at different rates, with coordinates from $E/2$ to $E$ being at the same rate as what the Transformer decides and coordinates from 0 to $E/2$ linearly changing from moving at the same rate to being the slowest. Allowing the embeddings to move at different speeds in every intermediate decoder layer, from the lowest possible speed to the original speeds, and to allow the attention mechanism to make use of multi-scale features moving at different rates at every layer and every token is a compelling idea as we see in the next section.

# 4 Experiments

In this section, we explain how we incorporated the idea of infusing wavelets into a large language model pre-training. We trained all of the models from scratch, which required substantial computing. However, the main aim of these experiments is to show how the performance of the models across three modalities improves with/without doing intermediate modifications on embeddings. Since we do not add any parameters when we modify intermediate embeddings with wavelet transform, we can compare the two models in terms of the performance boost the new architecture achieves and speedups.

## 4.1 Baseline And Training Setup

All models, similar to the GPT-2 architecture, consist of a stack of Transformer decoder layers. Since each requires pre-training the models from scratch, we choose the following setup. Every modality, namely text, symbolic music, and raw waveform, has the same architecture topology with a context length 512. We choose the number of decoder blocks to be 10, with 128 as the embedding dimension, the feed-forward dimension to be 512, and the number of heads to be 8. We opt for a two-layer feed-forward MLP inside the Transformer block after the attention block instead of a single layer typically used in Vaswani et al. (2017), with both the layers sharing the same number of neurons, i.e., 512, that of the feed-forward dimension. The final output layer of the Transformer decoder is then followed by a dense layer of 2048 neurons, followed by a dense layer of the same size as the vocabulary. This vocabulary size varies in the three modalities. For text8, it is 27, which is the number of characters plus an added extra token for space. For the raw waveform, we use an 8-bit resolution waveform at 16kHz, which is similar to the reported in (Goel et al., 2022; Verma, 2022), thus yielding 256 as a vocab size. For symbolic music, we utilize Google's tokenizer (Huang et al., 2019) to convert MIDI data to discrete tokens yielding 388-sized vocabulary. The baseline models in all three were simply a stack of Transformer decoder blocks without tinkering with any embeddings. For the proposed architecture we explained in the previous section, retain half of the embedding coordinates without any tweaks. For the other half, we impose a multi-scale structure parameterized by the coordinate in the embedding dimension for all intermediate layers. We do not add any single parameter in this setup and compare the performance with this tweak for all three modalities. We do this because we want to showcase the powerfulness of our algorithm for a rich variation of modalities for LLM pre-training. We do not compare against powerful, larger architectures going after scale, as this paper required pre-training from scratch. Instead, we take a shrunk-down version of GPT-2 architecture, viable in *academia with limited resources* and compare it with/without adding wavelets to the architecture regarding pre-training performance. All models were trained from scratch in the Tensorflow framework Abadi et al. (2016) for 25 epochs. We used a mirrored strategy for multi-GPU training. The learning rate schedule was chosen to be 3e-4 to start with reduced till 1e-5, whenever loss started plateauing. The number of training points available in all three models was 1M, yielding the total number of tokens to be 1/2 billion. These were randomly cropped from the dataset of choice. Apart from setting a default dropout rate of 0.1 in MLP and attention layers, no other regularization was done. The performance metric chosen to compare is only the negative log-likelihood loss, as this method improves the core architecture of the transformer-based GPT and helps achieve the objective we want to achieve: predict the next token. Since we are operating on intermediate embeddings, our work can hopefully generalize to setups with structured data similar to text, raw audio, and symbolic music, where one can go from a fine-grained structure to a coarse structure. We can see from Figure 3 how, for a toy example, we can impose a multi-scale structure that allows the attention mechanism to not only learn dependencies across various embeddings but also inject some information that can capture coarse and fine-grained structure into these embedding coordinates.

## 4.2 Performance on modalities

In this section, with the added modifications, we compare the performance of our baseline architecture across three modalities, namely text, symbolic music, and audio waveform with/without the addition of wavelet-based intermediate operation. We see that we substantially increased performance in all three modalities when we trained for the same number of training steps. To give an analogy for natural language, a decrease of 0.04 in validation loss is akin to going from a 16-layer architecture to a 64-layer model on

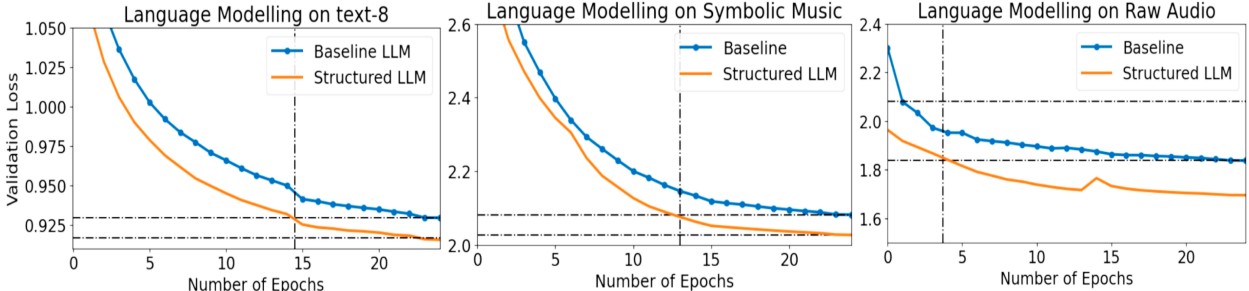

Figure 4: We report results for three modalities: natural language, symbolic music, and raw audio. We see that we achieve much faster performance than the baseline, almost twice as fast on shrunk down GPT like architecture. When trained for the same number of epochs, we see a substantial improvement in the pre-training performance, equivalent to a much larger architecture. The black vertical line denotes the epoch at which our architecture achieves the same performance as our baseline architecture.

a text-8 dataset (papers-with code, 2024). As shown in Figure 4, we achieve the same loss almost twice as fast as the original architecture regarding training steps for a shrunk down GPT architecture. This is particularly important as the GPT-like architecture can indeed take advantage of the structure that we imposed on half of the embedding dimensions. This speedup, i.e., the number of epochs/steps taken to achieve the same performance when the loss starts to plateau, is even smaller for raw audio. One of the reasons this can be attributed to is that audio signals remain quasi-stationary for smaller time scales, i.e., 20ms-30ms for harmonic sounds. For a sampling rate of 16KHz, a context length of 512 would correspond to 32ms, which may be one of the reasons that some of the coordinates nail down the contents of the context in fewer coordinates onto which we impose structure. The convergence happens much faster for raw waveform LLM setup than it is almost twice as fast in text-8 and symbolic music. We also compare our modifications' absolute clock run times in both learnable and non-learnable setups. We report the time it takes to complete one epoch relative to our baseline architecture. We see from Table 1 that our method is computational inexpensive, as the only operation carried out was simple averaging for the case of Haar wavelet or learning a single filter convolutional kernel with variable context lengths over various embedding dimensions.

Table 1: Comparison of the negative-log likelihood (NLL) scores (log base e) for our architecture with three modalities with/without adding wavelet-based hierarchical structure and for learnable wavelet transform.

| Modality | Baseline | Proposed | Same Performance Epoch | SpeedUp | Relative GPU Hours |
|---|---|---|---|---|---|
| Text | 0.93 | 0.915 | 14.5 epochs | 42% | 1.013 |
| Raw Audio | 1.84 | 1.7 | 3.7 epochs | 85% | 1.042 |
| Symbolic Music | 2.08 | 2.02 | 13 epochs | 48% | 1.059 |
| Text (Learnable) | 0.93 | 0.91 | 12.9 epochs | 46% | 1.094 |

### 4.3  Effect of Depth And Model Dimension

Here, we explore two variants of our architecture – what would happen if we reduce the model dimension from 128 to 32 and reduce the number of layers. We carry out all the experiments for text-8. We can see that for the variant where we reduce the model dimension to 32 for a 10-layer Transformer decoder architecture with eight heads, the model still retains faster performance as a baseline, almost twice as fast as seen in Figure 4, and achieves the performance without doing the modification (as seen as baseline) in around ten epochs. For the second experiment, we retain the exact architecture as proposed in our experiments reported in Table 1. However, we only have 6 Transformer Decoder layers, keeping the rest of the parameters the same (feed-forward dimension four times that of the model dimension, eight attention heads) to see the effect of depth. We see that the model continues to hold and again achieves the performance of the model trained for about 25 epochs almost twice as fast. Both of these experiments are shown in Figure 4.

### 4.4 Making multi-scale kernels learnable

As described in the previous section, we can see that by adding no parameters onto a Transformer decoder layers by imposing a multi-scale structure, we can make pre-training significantly faster on shrunk down GPT like architecture. In this experiment, we allow each of the kernels to be learnable. In the previous section, we defined the shape of the kernel as a Haar wavelet. We looked at the approximate coefficients of intermediate layer activations across all layers, with different resolutions occurring at different embedding dimensions. Now, in this experiment, we allow each kernel to be learnable. So now, instead of a Haar wavelet operation, we allow each kernel to be learnable for getting the *approximate signal* for various resolutions. Before, we were taking average to compute the *approximate signal* at a particular embedding dimension, which is convolutional with a kernel of length $L$ equal to $(1/L, 1/L, 1/L, 1/L...)$. In this experiment, we make the $L$ length kernel learnable from scratch, another way to compute the *approximate signal*. This simple operation for our base models only allows 0.02M (20k) extra parameters to the Transformer decoder architecture. Unlike the previous setup, which did not add any extra parameter, this further improves our performance from 40% to 46% faster speedup to get a similar baseline performance, as seen in Figure 4. This was carried out on the text-8 dataset. All results are reported on cross-entropy loss computed to the base e. This further validates our method and showcases further improvements and strength of our work.

## 5 Long Range Arena Benchmarks

We adapt our architecture to benchmark long-range arena (LRA) tasks Tay et al. (2021). It consists of various datasets that allow models to handle long-range prediction over sequence tasks over diverse domains, pushing the ability of Transformer architecture and other variants. We use three modalities: text, images, and mathematical expressions to test the model's ability to understand similarity, structure, and reasoning over extended contexts. We only use transformer-based architecture as reported recently by Liu et al. (2024). The other variants are state space architectures and hybrid models. For text, we carry out text classification on IMDb review dataset (Maas et al., 2011) on byte-level data with a context length of 2048 as input. The goal here is binary classification, which determines whether a movie has a positive or a negative review. For images, we use classification on CIFAR-10 as part of the image modality of LRA benchmarks. It is a pixel-level classification of the image that takes in as an input a sequence of pixels with values ranging from 0-255 with a length of 3072 and the output being one of the ten categories as the output. Finally, we benchmark on Long ListOps. It tests the capability of the architecture to understand hierarchically structured data in an extended context setup. As described in the LRA paper by Tay et al. (2021), "The dataset is comprised of sequences with a hierarchical structure and operators `MAX`, `MEAN`, `MEDIAN` and `SUM_MOD` that are enclosed by delimiters (brackets). An example (much shorter) sequence is as follows:

**INPUT:** `[MAX 4 3 [MIN 2 3] 1 0 [MEDIAN 1 5 8 9, 2]]` **OUTPUT:** 5

In our task, we use a version of ListOps of sequence lengths of up to `2K` to test the ability to reason hierarchically while handling long contexts. In the above example, the model needs to access all tokens and model the logical structure of the inputs to make a prediction. The task is a ten-way classification task and is considerably challenging." We use the setup provided by Khalitov et al. (2022) to extract the data and be uniform with other benchmarks. We experiment with almost the same architecture for all three modalities and only change the embedding matrix to account for different tokenizers and output categories. For a baseline, we use a 6-layer **Transformer decoder** only architecture, that is, **causal**, with a model dimension of 32 and a feed-forward dimension four times that of the embedding dimension. We take the last token of the sequence as an embedding that is extracted for classification, thus a 32-dimensional vector, which is then followed by a dense layer of 2048 neurons and a dense layer equal to the number of categories. The input is passed through an embedding layer that converts discrete tokens into a 32-dimensional vector. The input vocabulary of text, image, and list-ops is 256, 256, and 16, respectively. The context length is 2048, 3072, and 1999 tokens, respectively. The output categories are 2, 10, and 10, respectively. For our modified architecture, similar to how we described earlier, we introduce our waveletGPT module between every decoder layer. We retain half of the embedding dimensions as is. For the other half, we use non-learnable kernels, increasing the kernel size from 2,4,8 to 512 linearly for dimensions from 16 to 32, retaining the causality assumption. This introduces highways that hierarchically process the data at every embedding and every Transformer decoder

Table 2: Performance of predicting outcomes of list operations in the LRA (Tay et al. (2020b)) as reported in Liu et al. (2024). Bold indicates the best-performing model and underlines the second best. We use a baseline architecture for all three benchmarks, as reported in section 5, followed by modifying the intermediate embeddings with no parameter gain whatsoever by imposing a hierarchical structure. We do not report non-transformer or hybrid architectures.

| Transformer Based Attention Models | ListOps | Text | Image |
|---|---|---|---|
| Transformer (Vaswani et al., 2017) | 36.37 | 64.27 | 42.44 |
| Local Attention (Tay et al., 2020b) | 15.82 | 63.98 | 41.46 |
| Linear Trans. (Katharopoulos et al., 2020) | 16.13 | 65.90 | 42.34 |
| Linformer (Wang et al., 2020) | 35.70 | 53.94 | 38.56 |
| Sparse Transformer (Child et al., 2019) | 17.07 | 63.58 | 44.24 |
| Performer (Choromanski et al., 2021) | 18.01 | 65.40 | 42.77 |
| Sinkhorn Transformer (Tay et al., 2020a) | 33.67 | 61.20 | 41.23 |
| Longformer (Beltagy et al., 2020) | 35.63 | 64.02 | 40.83 |
| BigBird (Zaheer et al., 2020) | 36.05 | 64.02 | 40.83 |
| Luna-256 (Ma et al., 2021) | 37.25 | 65.78 | 47.86 |
| Reformer (Kitaev et al., 2020) | 37.27 | 56.10 | 38.07 |
| FNET (Lee-Thorp et al., 2022) Non-Causal | 37.27 | 56.10 | 38.07 |
| WavSPA – Ada Transformer (Zhuang et al., 2024) - Non-Causal | 55.40 | **81.60** | 55.58 |
| Ours (GPT Baseline With Classification Head) | 41.65 | 65.32 | 49.81 |
| **Ours (WaveletGPT With Classification Head)** | **57.5** | 66.38 | **59.81** |

layer without adding any parameter similar to what we carried out for LLM. As we can see from Table 2, we achieve non-trivial gains in all three modalities where even small gains are worth reporting. We outperform non-causal based methods, e.g., (Zhuang et al., 2024) significantly with almost 2% on ListOps and 4.5% with a much smaller architecture, shallower model(ours 32 dimensions, six layers vs. 128 dimensions, eight layers). Compared to a non-causal architecture FNet, we significantly outperformed all three LRA tasks, with 20% points on ListOps and Image and 10% on text. One of the biggest jumps is seen in the ListOps task, which requires modeling a hierarchical, tree-like structure of the math operations, which our model is best suited as motivated earlier. To the best of our knowledge, and reported by Liu et al. (2024), this achieves the best Performance of simple attention-based Transformer architecture on Long-Range arena tasks.

## 6 Conclusion and Future Work

We showcase the powerful incorporation of a core signal processing idea, namely wavelets, into large language model pre-training. By imposing a multi-scale structure onto every intermediate embedding, we see that we can achieve the same performance 40-60% faster, compared to the same baseline architecture, with the addition of no extra parameter. We achieve a substantial performance boost if we train for the same number of steps. Our method generalizes across three modalities: raw text, symbolic music, and raw audio, giving similar performance speedups. Several exciting directions can be explored in future work, including incorporating more advanced ideas from wavelets and multi-resolution signal processing onto large language models. It will be interesting to see how the model behaves for different variants of multi-scale structures.

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

# A Comparison With Exponential Moving Averages

We compare our method with Exponential Moving Averages (EMA) on the intermediate signals. This is widely used in time-series analysis for smoothening data, and it is another type of way that can modify intermediate signals. We proposed Haar wavelet, a multi-resolution kernel that can look at the input signal at various levels of scales depending on embedding dimension. We will now compare it against an EMA baseline and motivate where we differ and are similar to our proposed method.

## A.1 Background

Loosely speaking, instead of a moving average filter taking the mean of the signal, an EMA uses a different kernel, i.e., an exponential function. Meanwhile, a moving average kernel assigns equal weight to all time points. If we assume that the $x_i^l(t)$ signal of length is equal to context length after the $l^{th}$ layer with $t$ being the token index going from 0 to context length $L$ at embedding dimension $i$, we can define the modified exponential smoothed version of the signal $s_t$ as

$$s_0 = x_i^l(0) \quad s_t = \alpha x_i^l(t) + (1 - \alpha)s_{t-1}$$

Where $\alpha$ is the decay factor, it always satisfies $0 < \alpha < 1$. We can observe that for each of the tokens, depending on the decay factor $\alpha$, we assign weights to the more recent values over the past values. When $\alpha = 1$, the weightage given is only to the current observation, and when $\alpha = 0$, it is just flat and gives equal weightage. The differences with moving average filters are evident i.e., first, the moving average filter gives

equal weight to all of the values in a window to update the values of a particular window. Depending on the value of $\alpha$, an EMA filter gives an exponential weighted kernel. However, from the definition itself, an EMA filter, irrespective of the value of $\alpha$, is an Infinite-Impulse response (IIR) filter, whereas a moving average filter is a finite impulse response (FIR) filter. Therefore, for every value update at a particular location, the values of dependencies of the previous samples will never be zero and relatively small. One can see that these values can add up significantly for some values of $\alpha$ when we are predicting the next tokens at longer context lengths. Due to the nature of the IIR filter, the values are never zero. They are assigned values weighed depending on the previous observation as $1$, $1 - \alpha$, $(1 - \alpha)^2$, $(1 - \alpha)^3$,...

On the other hand, our proposed method includes wavelets composed mainly of FIR filters, including Haar or Daubechies. They are, therefore, only limited to a finite duration and can be adapted in multi-resolution setups with varied window lengths, as we have proposed in our paper. This allows us to have multi-scale information where we look at any signal at different resolutions with varied window lengths, with no contributions from components outside the desired window. (as we set the contribution from those values as 0). EMA, on the other hand, would still have some contribution from every component due to its recursive nature. One could also have a version similar to our method where one could vary $\alpha$ depending on the embedding dimension $i$. The update equations would now be a function of $i$, i.e.

$$s_0 = x_i^l(0) \quad s_t = \alpha_{(i)} x_i^l(t) + (1 - \alpha_{(i)}) s_{t-1}$$

This would introduce different dimensions decaying at different rates. Even with varying decay rates, because of the inherent nature of the IIR filter, we still give weightage to all values, which are never zero, unlike the FIR filter, which utilizes a window and gives no weightage to values outside the window.

Training all possible values of $\alpha$ is beyond our scope and resources. We, therefore, give the best equivalent of the EMA algorithm with our proposed method, as described in the next section.

## A.2   Experiments And Results

We retain our baseline architecture precisely the same for text-8. We train for a context length of 512 with the same setup reported in our baseline section and the same dataset, with the only tweak being taking the baseline architecture and adding an EMA layer to it. We choose the number of decoder blocks to be 10, with 128 as the embedding dimension, the feed-forward dimension to be 512, and the number of heads to be 8. We opt for a two-layer feed-forward MLP inside the Transformer block after the attention block instead of a single layer typically used in Vaswani et al. (2017), with both the layers sharing the same number of neurons, i.e., 512, that of the feed-forward dimension. The final output layer of the Transformer decoder is then followed by a dense layer of 2048 neurons, followed by a dense layer of the same size as the vocabulary. This vocabulary size varies in the three modalities. For text8, it is 27, which is the number of characters plus an added extra token for space. Similar to our proposed method, we experiment with keeping half of the embedding dimensions in all the layers the same without any modifications. For the other half of the embedding dimension after all layers, we carry out EMA on 1-D signals, as described in the previous section, with $\alpha$ varying from 0 to 1 linearly for embedding dimensions 64 to 128. We see a drop in performance compared to our baseline architecture and achieve an NLL score of 0.94. For comparison, our baseline trained on text-8 scored 0.93, with our proposed method being 0.915 and 0.91 for learnable and non-learnable cases, respectively.

## A.3   Discussion

There can be many reasons why EMA degrades performance. One of them can be tuning $\alpha$. There can be many possible choices, and tuning them for an expensive LLM pretraining is tough. Our proposed method, WaveletGPT, on the other hand, has a simple way of giving the weightage, which is grounded in signal processing and outperforms EMA smoothening. Further, in our learnable section, the architecture can learn the optimal **weights** in which, depending on the space spanned by the intermediate signals found inside LLM, it learns weights from scratch at different resolutions from the finest, i.e., window length 1 to the coarsest, i.e., window length as the context length 512.

