# OpenReview forum: "WaveletGPT: Wavelets Meet Large Language Models"
_TMLR — Rejected by TMLR_

### Review · Reviewer_CpSy · 2024-07-29

**Summary Of Contributions:**

The paper presents work to improve the convergence during pre-training of GPT-style large language model by replacing half the intermediate embeddings at each transformer layer with a wavelet transform (Haar filter) of the original embeddings, therefore making it easier for the Transformer to directly learn from dependencies at different time scales. Authors report experiments on three corpora (text, raw audio, and symbolic music) using the same basic architecture and show that the wavelet-enabled model reaches lower training (or validation?) loss faster than the baseline. Making the wavelet function trainable further improves results slightly. Given access to constrained (academic) resources, authors do not fully train their models to saturation, measure performance on downstream datasets, or perform experiments on large datasets.

**Audience:**

Yes

**Claims And Evidence:**

No

**Requested Changes:**

- Please address the above weaknesses. In particular, I would like to see experiments on more standard datasets (e.g. those used in WavSpA?) that show that the model is reasonably well trained when compared to other models
- Please clearly specify the datasets used -- I am not sure which data "wiki-8/ wiki-text8" is. Neither the Mikolov paper nor the original website seem to use that moniker and describe the size of the data used here, and its relation to data used in comparable work
- How does the work compare to FNet, https://arxiv.org/abs/2105.03824? Are the ideas related? Or not? A broader discussion of relevant and related work would be very interesting.

**Strengths And Weaknesses:**

Strengths
----

- The paper is generally well written and easy to follow
- Training of LLMs is an important research area and deserves further study. Pre-training is the most cost-intensive part, so any improvements here will immediately benefit the larger research community
- Authors experiment with three different datasets from different modalities, which strengthens the validity of their claims

Weaknesses
----

- The paper does not analyze if the models have been fully trained, when training actually saturates, and how pre-training affects downstream performance. While I understand that pre-training behavior can be indicative of the models modeling power, the authors' claims would be much stronger if they could show that the model does not just converge faster (which I am also not sure they have shown, given the shape of the curves in Figure 3), but that downstream performance improves as well
- I am wondering how the proposed work compares to "WavSpA: Wavelet Space Attention for Boosting Transformers’ Long Sequence Learning Ability", Zhuang et al., 2023, arXiv:2210.01989v3 or "Structural Guidance for Transformer Language Models", Qian et al., 2021, https://arxiv.org/abs/2108.00104
- The authors perform very limited ablation studies and there are no comparable numbers with other methods available - how fast do other methods train/ converge? What is the performance on downstream tasks?

---

> ### Author Response · Authors · 2024-08-23
> **Response to reviewer CpSy Part 1**
>
> We thank the reviewer for all of the helpful comments and feedback. We have incorporated and addressed most of the points and carried out additional experiments to further prove the powerfulness of our proposed method.
>
> Question -- Authors report experiments on three corpora ….
>
> Author Reply: The paper clearly mentions that our architecture reaches a lower validation loss in Figure 3. There is no section or line in the paper mentioning the training loss. It is unclear why training loss is mentioned in the reviewer's comment.  The primary reason for reporting the validation loss is that it is well documented that scaling neural architecture (https://arxiv.org/abs/2001.08361) in terms of parameters improves model performance, keeping other aspects of the model the same, i.e., keeping the data, training recipe, loss function, optimizer to get the foundational model. We aim to use the same number of parameters while doing functional modifications to intermediate Transformer embeddings to improve that setup's performance substantially. Validation loss is one of the most important indicators of how well/good a foundational architecture is. In fact, in the Gemini technical report (https://arxiv.org/pdf/2312.11805), Figure 4 does not give the likelihood scores for Gemini Pro and Gemini Ultra architectures and intentionally omits labeling it.
>
> Comment: Given access to constrained ….
>
> Author Reply: Transformer decoder architectures are ubiquitous and widely used in open-source models such as LLAMA and are at the core of the modern AI uprising across modalities. LLAMA architecture based on Transformer decoders reported loss curves that seem to be decreasing even faster than our reported loss curves (https://arxiv.org/pdf/2307.09288) in Figure 5.
>
> The validation loss curves in all three modalities gave minor improvements after 25 epochs; thus, the training was stopped. Each epoch takes substantial computing time.  We ran the architecture for 25 epochs, with one epoch consisting of about 512 million tokens for text. The dataset initially for text consists of 100M tokens. We take random chunks so that every token can have different contexts for prediction during training.  This paper is only concerned about pretraining, which aligns with other papers exploring pretraining ideas. For example, "Character-Level Language Modeling with Deeper Self-Attention" https://arxiv.org/pdf/1808.04444only reported bits per character performance, a likelihood loss function, without reporting on other metrics. MegaByte(https://arxiv.org/abs/2305.07185) only reports on likelihood scores and other variants. It is widely reported that lowering the negative likelihood score or perplexity will yield better downstream results if all other aspects of the model are kept constant (same data, context, loss, training steps/same fine-tuning/RLHF, etc.)
>
> So this work focuses on not chasing the size/number of parameters but keeping all other blocks constant, improving the core architecture. Other papers, too, e.g., Music Transformer in Symbolic Music, primarily report for NLL score improvement throughout. The author knows that the model is far from state of the art as it is a shrunk-down version of GPT-2/GPT-3.
>
> The paper has added a section on downstream tasks and benchmarked against 3 Long Range Arena benchmarks that were appropriate for this paper, namely text, image, and ListOps, achieving state-of-the-art results for Transformer methods. Here, too, the author did not add any parameter to the architecture, and the gains were achieved by imposing a multiscale hierarchical structure. We also have added a section on how depth and model dimension affect our work.
>
> Perform experiments on large datasets: We have performed experiments on large datasets that were best suited to a training budget in academia. Text-8 contains 100M characters and is widely reported. Similarly, the other two datasets, YouTube Mix-8 and MAESTRO, have been used for research purposes.
>
>
> Comments: Training of LLMs ….
>
> Author Reply: We agree that pretraining is the most cost-intensive part of pre-training a large language model. The paper shows we can achieve the same pre training performance almost twice as fast with a simple recipe across three modalities. This idea is grounded in signal processing, and we intend to release a simple tool/function that can do this across any Transformer decoder block upon acceptance. This is quite powerful as with a simple function call, it will help improve pre training performance and further significantly boost it. We have benchmarked it on three datasets to show how this method can be applied to different modalities, including how ubiquitous Transformer architectures are. This is the reason we applied our method to three modalities. We have added absolute time speedups in terms of GPU hours, further strengthening our claim and giving evidence.

---

> ### Author Response · Authors · 2024-08-23
> **Response to reviewer CpSy Part 2**
>
> Comment: The paper does not analyze ….
>
> Author Reply: We beg to differ from this. We can see that for all three modalities, the validation loss starts plateauing and gives minuscule gains. As described with some of the references, this paper cares about the pretraining performance and not about downstream tasks that are a function of the model size and the training corpus. We have reported on similar lines the negative log-likelihood performance similar to the Music Transformer paper, MegaByte, and Character Level Attention Model work …. We have added a section on the downstream tasks on one of the most popular benchmarks -- Long Range Arena --. It boosts the performance of Transformer based architecture to the state of the art accuracy compared to other methods showcasing the power of our modifications.
>
> Comment: I am wondering how the proposed work compares to "WavSpA:.....
>
> Author Reply: The author has added a section in the introduction to compare our method with WavSpA and FNet.  In short, it is an efficient method for advancing the attention block, with attention being carried out in wavelet transform space instead. They take the wavelet transform, compute the attention, and then take the reconstruction back to advance the Attention Block. However, this cannot be applied to causal architectures such as GPT or LLM setups. Figure 1 of the paper looks at future tokens to modify the current tokens. This is at the heart of computing wavelet Transforms. For FNet, it replaces the attention block with 2-D FFT, which is an idea similar to WavSPA. This looks into future tokens to modify the current tokens and is well-suited for BERT-like architectures. The structural guidance paper is only using domain knowledge to condition LLM pre-training whereas we do not use any knowledge and show a single algorithm working for multiple domains.
>
> Comment The authors perform very limited ablation ….
>
> Author Reply: We have performed ablation studies concerning depth and model dimension. We have also added absolute training time benchmarks. We have added a section on Long Range Arena Benchmarks. We have compared our baseline and the proposed modifications with those of other transformer-based attention architectures. We do not compare against state space architectures or Hybrid methods as it will be an unfair comparison.
>
> Requested Changes:
> Please address the above weaknesses. In particular, I would like to see experiments on more standard datasets (e.g., those used in WavSpA?) that show that the model is reasonably well-trained.
>
> Author Reply: We have added more experiments to compare against standard benchmarks on Long Range Arena benchmarks. We have achieved substantial non-trivial gains on Transformer-Based Attention Models. We cannot compare our method with that of WavSpA, as described in the previous section. It uses non-causal operations. There is no end to running larger, deeper models. However, the goal is to take a baseline architecture and improve its performance and convergence speeds. We have, however, compared it against our GPT baseline and WaveletGPT for the LRA benchmark by using a causal architecture with a much smaller, shallower architecture than WavSPA.
>
> Comment: Please clearly specify the datasets…..
>
> Author Reply: We have clarified and updated the manuscript accordingly. We use the text-8 dataset, which consists of 100M characters from Wikipedia.
>
> Comment: How does the work compare to FNet…..
>
> Author Reply: We have added this in the related work section.  Our work is not at all related to it. FNet proposes replacing the attention block with that of computing a 2-D Fourier Transform and replacing it with two 1-D FFTs. It can only be applied to BERT-like architectures, as to take Fourier Transform, the entire signal is considered. This is not interpretable, and the authors also do not give specific reasons for it, and no signal processing operation takes an FFT 12 times. It works well and achieves BERT-like performance efficiently, and that matters!  Our work is designed specially for causal LLM architectures like GPT and only modifies the intermediate embedding by doing simple algebraic operations which are causal in nature.
>
> Broader Impact Concerns:
> n/a
> Claims And Evidence: No
>
> Author Reply: The paper claims that we improve the convergence times without adding any parameters. Further retaining the same number of parameters, we increase the pretraining performance across three modalities. We have added downstream benchmarks on Long Range Arena tasks to achieve state-of-the-art performance on transformer-based architectures and their variants. These are our claims, and we have provided evidence to prove them.
> Audience: Yes

---

### Review · Reviewer_Zdii · 2024-08-03

**Summary Of Contributions:**

The work is targeting improving the efficiency for pre-training phase of LLMs, so that the same loss can be achieved in smaller number of steps while not introducing any new parameters. Authors propose to do this by transforming every output of the decoder layer into new representation (not learnable + fast). This transformation keep half of the embedding the same, while the the other half is considered as D/2 1D signals in time (time is equal to the context total size). Every signal then transformed with the window averaging where every window size is different per embedding dim + keeping the causality property of the model. Author makes the connection / derivation for the Haar wavelets so that every dimension in the intermediate representation is considered as 1D signal and only particular signal resolution is preserved by applied wavelet transformation. Author demonstrate idea on text (wiki-8, char based LM), audio and music data with GPT-2 model showing that they can speedup the loss convergence by almost 2x.

**Audience:**

Yes

**Broader Impact Concerns:**

The work is targeting to improve efficiency of LLMs, which should have positive effect on the compute and broader accessibility of LLMs to the rest of the world as well as less energy consumption.

**Claims And Evidence:**

No

**Requested Changes:**

**Text itself:**
- page 1 "has been show" -> "has been shown"
- reference style - use citet, citep where needed + spaces before reference starts
- page 1 "GPT3 are Brown" - put reference at the end, not between the verbs
- page 1: why reference only to Gemini?
- page 1: "From a few that are listed now" - not clear for me what you mean with it
- page 1 knowledge distillation - good to have link to Hinton, et al paper.
- related works - how about https://arxiv.org/abs/2205.13147 - or similar direction works? There are works also on diffusion where people use multi-scale representations. Let me know if you need more references on that.
- add prior works with similar models on the considered data - yep, you are not trying to beat them, but the baselines you use should be reasonable for the data / model size you use. Right now the loss for me is too high (even for the wiki-8 char based model)
- Fig 1 caption "We find" - why "find"? also "word" -> "world".
- page 3 "as we did in prior works" -> you did? or was done by other people?
- for notation - for layer index better to use $n$ as you have $N$ layers. For time you name it $L$ which creates confusion with layer index. Better also to use $T$ for the context.
- You do math, but then do not give definitions for $h$ and $g$
- page 4 "we will ... will also" - English is incorrect
- How $c_j$ are defined on page 6?
- All description in 3.3 section - two last paragraphs - why not put the formulas? This is exactly your algorithm. You give formulas for the wavelets intro, but no clear definition for your method.
- Sec 3.4 paragraphs 2-3 are too long with high repetition on the meaning for the same things.
- Page 7 notation is weird to me $IX$, $xn$.
- Sec 3.4 Why do you discuss a lot approximate signal and wavelets, but later in formula here you don't have anything of that and just use the signal itself?
- "We opt for a two layer" on page 8 - what does it mean? Also can you point how many parameters you have in your model?
- page 9 "increase of 0.04" -> decrease?

**Experiments:**
- a lot of things I listed in the weaknesses section which should be provided in the paper
- also you mention that you train till convergence, but I do not see plateau in Figs 3 tbh.
- one more thing - people now do interleaving the conv with self-attention (e.g. conformers in speech domain) - how your method then is different from that if you do simple window averaging?

**Strengths And Weaknesses:**

**Strengths**
- Idea itself on interpreting the intermediate representations and transforming to particular stat of the signals
- Shown speedups (if I assume empirical part is strong and convincing)

**Weaknesses**

Exposition
- long and unnecessary introduction to wavelets
- final algorithm does not seem to require this long introduction to wavelets, or even reference: simple window averaging with no connection really to wavelets -- this makes reading is really hard as I tried to connect and infer the final algorithm for long time from the introduction, and could not follow the math at all
- no clear formulation of the final method -- there is mentioning about causality, but clear math formulation or pseudo-code is absent
- performance-wise final algorithm does not seem to benefit from efficient implementation of wavelet transforms - so why then discuss them half of the paper?
- there are several places with weird English sentences

Alternative approaches
- final modification seems like a channel-wise window averaging (which is only a part of wavelet transform). Similar, but more efficient technique (exponential moving average) is widely used in autoregressive modeling (see QRNN, State Space Machines), so that's mandatory to compare against then.

Comparison
- no analysis of time/memory efficiency for proposed modification (which will be beneficial for the paper as it should not introduce any overhead. Authors discuss the parameters overhead, but not the forward / backward time overhead for the transformation)
- ungrounded choice of problem for benchmarking: wavelets are technique for the continuous signals, so speech and music processing will be good applications. But how this can be useful for text and text representations where it is discrete signal (and this is main problem e.g. for applicability of diffusion models to text inputs with different new techniques how we can do it).
- no ablations on the different model size (only one size is considered - I am on the authors side about academia limit on the compute, but at least try smaller model size to check that there is something consistent)
- non standard baselines to compare with for the chosen model - e.g. I don't see that wiki-8 with char based LM is used anywhere for benchmarking. Also it is not clear if loss of this modified model will translate well into downstream task.
- experiments are done with many epochs over data, but if we emulate LLMs - we do 1 epoch over data there, so it is really not practical comparison then.
- no deep ablations on what transformations we should take:
  - why only average and no difference too as in wavelets?
  - why apply only one average to every embedding dimension? why not taking several statistics of the signal? or how at least this affects the training? Here you kind of make assumption that all signals are disentangled which may not be true. I really don't get why for different dimension you take different level of signal downsampling only (no difference, and no other info). With this procedure, I would expect model to be weaker, as you loose a lot of info, the only way it is helpful if it is redundant info and you are at the long range dependencies / long sequences, which you really didn't test I think.
  - what changes for different embedding sizes? So if I will train LLM where the embedding size will be increased significantly, what should I do then?
  - how do we depend on the depth of the models? do we do transformation after every layer? or maybe it should be done only at the beginning / at the end of the NN?

---

> ### Author Response · Authors · 2024-08-24
> **Response to Reviewer Zdii Part 1**
>
> Summary Of Contributions:
> The work is targeting improving the efficiency for pre-training phase of LLMs, so that the same loss can be achieved in smaller number of steps while not introducing any new parameters. .....
> Idea itself on interpreting the intermediate representations and transforming to particular stat of the signals
> Shown speedups (if I assume empirical part is strong and convincing)
>
> Author Response: We thank the reviewer for all of the helpful comments and feedback. We have incorporated and addressed most of the points and conducted additional experiments to prove our proposed methodology's power further.
>
> Author Reply:  We agree that interpreting the intermediate representations and transforming them to a particular embedding space of interest is exciting and needs further exploration. We have further strengthened the empirical part by benchmarking for three long-range arena tasks and achieving state-of-the-art results for Transformer based attention architectures without replacing or modifying the internal block of the Transformers.
>
> Weaknesses
>
> Exposition
> == long and unnecessary introduction to wavelets
>
> Author reply: This is a machine learning venue where some of the machine learning readers are getting introduced to wavelets for the first time. Given the venue, this was required. I would have explained the ideas more succinctly for signal processing venues such as ICASSP/ Signal Processing Letters.
>
>
> == final algorithm does not seem to require this long introduction to wavelets, or even reference: simple window averaging with no connection really to wavelets -- this makes reading is really hard as I tried to connect and infer the final algorithm for long time from the introduction, and could not follow the math at all
>
> Author Reply: The paper does require this introduction. The other reviewer mentions that the paper is easy to follow. Nevertheless, we have tried to modify the paper in several places. This is a machine learning venue with readers sometimes hearing the term wavelet for the first time. To say "window averaging has no connection to wavelets" is incorrect. Almost all wavelet textbooks, and the tutorial we have referred to mentions that the Haar wavelet is only averaging. Averaging is a convolution operation  :) And convolutions are at the heart of wavelet definition.
>
> As you can see from the definition, wavelet consists of convolutional operations. So, with the same analogy no one says that wavelets are just convolutions. We made the kernels learnable in the section, which improved our model performance even further. By then, it is no longer a simple window average learning far more complex structure at every layer at every embedding dimension with variable kernel length mimicking a wavelet filter bank :). So yes, it is a straightforward window averaging only on one variant method, but saying it has no connection to wavelets would not be appropriate. We take averages (and learn kernels) across multiple time/token scales, and this is at the heart of multi-rate signal processing and the idea of wavelets.
>
> == no clear formulation of the final method -- there is mentioning about causality, but clear math formulation or pseudo-code is absent
> Author Reply: We have added a pseudo-algorithm description. We intend to open-source our model architecture upon acceptance of our paper. We have also described the work in section 3.4 more clearly now.
>
>
> == performance-wise final algorithm does not seem to benefit from efficient implementation of wavelet transforms - so why then discuss them half of the paper?
> Author Reply: The paper itself is based on wavelet decomposition and adapts it causally. The similarities are easy to see. Once we make the kernels learnable with variable kernel lengths, the idea is the same as taking a wavelet decomposition at various levels, with the only difference now that we are learning the type of wavelet decomposition from scratch.
>
> == there are several places with weird English sentences
> Author Reply: We have tried to articulate the method to the best of our ability. It has been improved in several places now. The other reviewer mentions, "The paper is well written."

---

> ### Author Response · Authors · 2024-08-24
> **Response to Reviewer Zdii Part 2**
>
> Alternative approaches
> == The final modification is a channel-wise window averaging (which is only a part of wavelet transform). A similar but more efficient technique (exponential moving average) is widely used in autoregressive modeling (see QRNN, State Space Machines), so it's mandatory to compare them.
>
> Author Reply: We do not carry out an exponential moving average, and the method is quite different. Even if we were to compare against the exponential moving average, we take averages at multiple scales instead of fixed moving average operations with fixed weights. The method that we have proposed finally learns the kernels or convolutional weights from scratch at multiple time scales, which can be thought of as learning different weighting functions at different kernel lengths and scales from scratch optimized for the next token prediction. To our knowledge, we have yet to see such a framework anywhere in the literature.  Conformers are the closest, but it takes multiple filters to look at all the embedding dimensions with the same resolution filters. We treat each embedding dimension separately with a single filter (learned or fixed) across the 1-D signal dimension.
>
> QRNN architectures are from the pre-Transformers era, which uses convolutional and pooling layers. On the other hand, we use Haar wavelet or learnable wavelet operation as part of post-processing the output of the Transformer Decoder output. Since wavelet operation is nothing but convolution, it does look similar. We do not use pooling layers. We have compared it with other Transformer methods in the Long Range Arena benchmark. To compare against State Space Machines would not be fair as it is a different category of algorithms. It was fair to compare with the same architecture with/without the addition of modification to have complete control. Several aspects, like model dimension/layers/ training recipes, etc., are different and make it difficult to compare fairly. We also share that the increase in performance on text-8 by reducing the loss by 0.04 is akin to going from 16-layer to 64-layer architecture. Further, the gain is similar for the case of symbolic music as well if we compare the decrease in loss with that reported in the Music Transformer paper.
>
> Comparison
> == There is no analysis of time/memory efficiency for the proposed modification (which will be beneficial for the paper as it should not introduce any overhead. Authors discuss the parameters overhead, but not the forward/backward time overhead for the transformation)
>
> Author Reply: We do not add much memory overhead and have added the absolute time complexity in Table 1. This shows that we achieve a significant performance boost with little to no change in computational complexity.
>
> == Ungrounded choice of problem for benchmarking: wavelets are a technique for continuous signals, so speech and music processing will be suitable applications. But how can this be useful for text and text representations where it is a discrete signal (and this is the main problem, e.g., for the applicability of diffusion models to text inputs with different new techniques we do)?
>
> Author Reply: This comment is confusing, and we are unsure how to respond. We agree that wavelets are techniques for continuous signals, so speech and music processing will have good applications. Text and text representations are discrete only for input (tokenizer or character-level representation). After that, once they are converted to embeddings, the representations are, in fact, continuous. Hence, the intermediate embeddings after Transformer decoders are all continuous signals; thus, we can apply wavelets to them.
>
> == There are no ablations on the different model sizes (only one size is considered - I am on the author's side about academic limit on the compute, but at least try a smaller model size to check that there is something consistent)
>
> Author Reply: We have added ablations with two more experiments - -depth reducing it from 10 layers to 6 layers and a smaller model that reduces the model dimension from 128 to 32. For both of these, we retain our hypothesis that it converges faster and improves the architecture's performance.
>
> == Nonstandard baselines   I don't see that wiki-8 with char-based LM is used  for benchmarking.  if this modified model's loss will translate well into downstream tasks.
>
> Author Reply: Our networks are too small to compare with the downstream tasks. Ten layers with a model dimension of 128, 8 attention heads as opposed to a typical LLM trained from the industry having 24-100 Layers with hidden dimensions 1024-3072 and attention heads from 16-24. We can only compare what we get before and after modifications/pre-training with what we get before and after modifications/pre-training. We have compared the relative gains and given the reference of the paper with the code in the paper. Wiki-8 was a typo and has been corrected to text-8, a clean version of enwiki8 everywhere now.

---

> ### Author Response · Authors · 2024-08-24
> **Response to Reviewer Zdii Part 3**
>
> == Experiments are done with many epochs over data, but if we emulate LLMs - we do one epoch over data there, so it is not a practical comparison.
>
> Author Reply- With due respect, LLM training is typically carried out over epochs or training steps for smaller datasets. This is well documented here: https://developer.nvidia.com/blog/training-bert-with-gpus/. Further MegaTron-LM paper from NVIDIA https://arxiv.org/pdf/1909.08053 in Section 5.2 for GPT2 pre-training over WebText mentions a simple relationship of epoch with iterations and how the training was carried out over multiple epochs.
>
> == No deep ablations on what transformations we should take:
> Why only average and no difference, too, as in wavelets?
>
> Author Reply: We have added this to the paper and explained it here. The approximate coefficient captures data at multiple scales, and this is what we intend to capture. Data around has a multiscale structure that approximate coefficients of wavelets can approximate. So, in text, the data is letters, words, paragraphs, and topic models. Similarly, this is true for music, where the data goes from notes to motifs to sections to whole pieces. This hierarchy is captured by approximate coefficients only and is interpretable. In the paper, we mentioned that it would be challenging to incorporate all of the wavelet approximations into the architecture without significantly altering the transformer structure. We, hence, do not incorporate difference-based and go along a single branch of a wavelet tree.
>
> == Why apply only one average to every embedding dimension? Why not take several statistics of the signal? Or how does this affect the training? Here, you assume that all signals are disentangled, which may not be accurate. I don't get why. For the different dimensions, you only take different levels of signal downsampling (no difference and no other info). With this procedure, I would expect the model to be weaker, as you lose a lot of information; the only way it is helpful is if it is redundant info and you are at the long-range dependencies / long sequences, which you didn't test, I think.
>
> Author Reply: Averaging has a specific meaning in this regard as it is tied to Haar wavelet. It also serves the purpose of improving every latent representation in the Transformer based LLM model by having hierarchical information at multiple time scales. It will be very difficult to incorporate all of the statistics without major alternation in the Transformer architecture. We do not assume that signals are disentangled. The network learns a representation that incorporates dependencies while incorporating hierarchical information in different coordinates of each embedding in every Transformer decoder layer. We do not lose information, as we can see from all of the experimental evidence; in fact, it is contrary. There is also no definition of long-term dependency. We have benchmarked our architecture in the Long Range Arena benchmark, which has quite a long context, and our method shines and performs well with the addition of no extra parameters reinforcing our claims with evidence.
>
> == What changes for different embedding sizes? So, if I train LLM where the embedding size will be increased significantly, what should I do then?
>
> Author Reply: We cannot run for every hyper-parameter. However, the algorithm explanation is clear, and there is no bias towards a particular embedding dimension. We cannot run for large embedding dimensions, but we have experimented with them. We have added an algorithm description in the paper that clearly outlines how to modify the intermediate embeddings when we have different-sized embedding dimensions.
>
> == How do we depend on the depth of the models? Do we do transformation after every layer? or maybe it should be done only at the beginning / at the end of the NN? We have added an experiment on the depth of the architecture. We do not make any assumptions about the depth since we only modify the output of the Transformer decoder.
>
> Author Reply: We have added an experiment on depth. We see no reason for depth to be a concern as, in some ways, we improve the core Transformer model. Yes, we do transformation after every layer. In some ways, this computes embeddings implicitly, retaining a multiscale structure where the coordinates of every embedding have information moving at different rates. Combining this with multi-head attention is a powerful idea. Yes, the reviewer is correct in that we are not making any assumptions about the depth since we are only modifying the output of the Transformer Decoder.

---

> ### Author Response · Authors · 2024-08-24
> **Response to Reviewer Zdii Part 4**
>
> Requested Changes
>
> Most of them have been done. We will add explanation to some of them.
>
> == related works - how about https://arxiv.org/abs/2205.13147 - or similar direction works? There are works also on diffusion where people use multiscale representations. Let me know if you need more references on that.
>
> Author response - I have added a note to the paper. This is different. It chops off the classification embedding vector into d, d/2, d/4, and d/8 dimensional embedding, and so on, and operates the loss function on all the intermediate representations instead of just the d dimensional vector. This is only carried on the last layer. Our paper is a much more complex version as it embeds every intermediate embedding with the multiscale version in all the Transformer decoder layers. So, for a d-dimension vector, a coordinate will move at only the coarsest scale and is the same for all the tokens in the context at a particular layer. Then, some coordinates are the same for half the tokens across context length, and so on to impose a tree like structure. For a d dimensional vector, d/2 coordinates are left unchanged, whereas for other d/2, a multiscale structure is imposed. Unlike Matryoshka Representation Learning, there is no special loss function at the end to guide the hierarchy. Still, the hierarchy is implicitly imposed driven by the  next token prediction criteria on all of the embeddings present across all layers in an LLM.
>
>
> == Add prior works with similar models on the considered data - yep, you are not trying to beat them, but the baselines you use should be reasonable for the data/model size you use. Right now, the loss for me is too high (even for the wiki-8 char-based model)
>
>  We have reported the results on a context length of 512. Hence, it might be too high, as the validation loss is a function of data, context length, and model size. There is little difference between the large GPT-2/GPT-3 architecture and the architecture we have trained, except scaling up on model dimension, number of heads, and number of layers. It is far beyond anyone in academia to train such an architecture. We show comparable numbers for text-8, and according to scaling laws, these numbers will keep improving. We have to draw the line somewhere as to what we chose to work with 128 as the model dimension with eight heads and ten layers with a context length of 512. During training, the model saw a total of 25 epochs times 0.5 billion tokens, which is substantial. Our model, even smaller than on the Long Range Arena benchmark, is doing much better on ListOps and Image tasks.
>
> == Sec 3.4 Why do you discuss a lot of approximate signals and wavelets, but later in the formula here, you don't have anything of that and just use the signal itself?
> Author Reply: We have put up an algorithm that explains this in more detail. For half of the embeddings, we use the embeddings as is. For the other half, we take the approximate coefficient approximation via either Haar wavelet, which is an averaging operation, or learn the kernels from scratch. We have put up the algorithm, and it is self-explanatory.
>
> == Lot of things I listed in the weaknesses section which should be provided in the paper
> Author Reply: We have done that to the best of our ability, run several additional experiments, added the algorithm explicitly, and benchmarked on long-range arena tasks.
>
> == Also, you mention that you train till convergence, but I do not see a plateau in Figs 3 tbh
> Author Reply: We can see from the plot that for 1 epoch, the architecture started to give minuscule gains (taking 500 million tokens per epoch). The rate of decrease has definately started to reduce. Further, we can see that from papers like LLAMA, two loss curves with an even rate of decrease have been reported. We have changed the wording of plateau now.
>
> == People now interleave the conv with self-attention (e.g., conformers in the speech domain) - how is your method different from that if you do simple window averaging?
>
> Author Reply: Yes, people are interleaving convolution operations with self-attention. It is different from the definition of wavelet decomposition we have described. Simple window averaging is only one variant. Averaging again is a convolution operation.  One would argue that convolution operator is closely related to weighted moving averages operator. And Wavelet decomposition technique is carried out using convolutions, and we have tried to do that. Conformer architectures usually learn filters across all embedding coordinates with multiple filters with a single resolution. On the other hand, we learn a single convolutional kernel per embedding coordinate with multiple scales for different coordinates, mimicking a wavelet decomposition using convolutional filters or averaging operations.

---

> > ### Comment · Reviewer_Zdii · 2024-09-05
> > **Some comments on the authors new info**
> >
> > Dear authors,
> >
> > I am in the progress of reading your revision. But so far some comments on your replies:
> >
> > > Regarding the exposition:
> >
> > I agree with your point it is ML journal, so some math intro is helpful. However you do deeper into to wavelets and then it is hard to connect simple averaging (yep, Haar wavelet, convolution) to the general wavelet theory you are talking about. In case I understand the wavelets (which I do) I was trying to see something more advance in the method, so it is confusing in the sense of giving more general math but then not using it all, and just following simplest wavelet. I would like to see then more intuition explanation first and then connection to math / potential extension to more complicated wavelet settings which are doing not only averaging as Haar case.
> >
> > > By then, it is no longer a simple window average learning far more complex structure at every layer at every embedding dimension with variable kernel length mimicking a wavelet filter bank :).
> > > The similarities are easy to see. Once we make the kernels learnable with variable kernel lengths, the idea is the same as taking a wavelet decomposition at various levels, with the only difference now that we are learning the type of wavelet decomposition from scratch.
> >
> > First, I see that learnable is not bringing much improvement really. So this questions why I need to have learnable kernels in practice? Second, you don’t really show which type of the wavelet that we learnt and it could be that properties are not good. Even can we say that these learned kernels give us the wavelets? In this case it is not clear why to go for the wavelets besides the nice connection and intuition only.
> >
> > > The other reviewer mentions, "The paper is well written."
> >
> > Yep, it is overall well written, but if I go deeper into math and strictly following your sentences - I don’t get many places. Sorry, I was super strict as I interpret this also partially as a math paper as soon as you started to discuss math and make connections.
> >
> > > Re: Alternative approaches
> >
> > I think you didn’t get my point. Point was that you introduce something new (and yep, I agree exactly this thing was not done before) but you don’t compare with any obvious baselines, e.g. doing EMA computation with different decay factor, interleaving with convs, aka averaging in SSM, etc. So I don't get why I need to have such configuration as you propose as no grounding why other simpler things and even things people already doing can be just reused? Even you tell me that you are trying to avoid the transformer model change, I could argue that SSM converges faster w/o loss of the performance now then why should I use your model instead of SSM or SSM+self-attention? why your solution is better? Either you show it is better and simpler or it is better, or some trade off between simpler/practical vs better/worse. I think some baselines are missing from the paper to make a decision for reader why to switch to your method.
> >
> > > Author Reply: We do not add much memory overhead and have added the absolute time complexity in Table 1. This shows that we achieve a significant performance boost with little to no change in computational complexity.
> >
> > Here the relative GPU hours is per step? or for the total training time?
> >
> > > After that, once they are converted to embeddings, the representations are, in fact, continuous. Hence, the intermediate embeddings after Transformer decoders are all continuous signals; thus, we can apply wavelets to them.
> >
> > What I meant is that then your embeddings and overall operations you are doing are not interpretable in some sense. You could argue that we build some continuous space for language where some continuous operations are valid, but it is not clear how you interpret them. I would say that having results for continuous domain will strengthen the paper, as first you show that for the data of proper nature it is working, and then you show that in text it is also working + you could say that the embedding space we build is with the proper properties we are searching for (though some analysis of this could be needed).
> >
> > > Re: Nonstandard baselines
> >
> > I am ok to compare just with the baseline model you have, w/o going to LLM SOTA models both on NLL and downstream tasks. However main concern is having char-based and not word-piece/word based models.
> >
> > > We, hence, do not incorporate difference-based and go along a single branch of a wavelet tree.
> >
> > Then I still don’t get how this is different from other works on multi-resolution things, e.g. conformer, or with different pooling except the way how exactly you do the averaging (convolution).

---

> ### Author Response · Authors · 2024-09-17
> **EMA + other concerns**
>
> Dear Reviewer Zdii and AE,
>
> Here are some of the main concerns addressed. We were expecting more comments as the reviewer mentioned that these are the comments for now and were waiting to address back our feedback.
>
> == We have compared our work against an EMA baseline using the same idea you suggested. Make the decaying factor vary from 0 to 1 for different embedding dimensions. It performs worse than the baseline, and I have added all the comments about our experiment in the appendix.
>
> == hardly any paper has been accepted at any conference on conformer pretraining for LLM. We differ from a standard speech-based LLM in multiple ways
> : i) The convolutions are carried over the embeddings and not across 1-D signals.
> ii) We learn multi-resolution kernels from the lowest to the highest window size across 1-D signals.
> iii) The convolutional filters learned are only 1 per 1-D signal, unlike a typical learning formulation, say 128 or 256 signals.
> iv) All of the operations are causal.
>
> == The proposed numbers are much stronger as they hold for four datasets. We see no reason why the method should not work for GPT-2 tokenization instead of letter-based tokens. The MIDI tokenizer was more complex than subword tokens used as it had pitch, duration, and energy information. A standard LLM recipe works across robotics, proteins, speech, images, music, audio, and text.
> We have given references to several papers that do character-based benchmarks.
>
> == Here are the relative GPU hours per step. Or for the total training time? Both are the same as the total training time; there are thousands of steps, so it is a simple multiplicative factor.

---

### Review · Reviewer_t874 · 2024-08-08

**Summary Of Contributions:**

This paper proposes imposing a wavelet structure for a transformer-based auto-regressive model, a widely used framework in LLMs. The method was applied to text modeling, raw waveform modeling, and music MIDI modeling. The paper focuses on the academic computing scale and shows the performance improvement from the vanilla transformer in terms of the convergence speed while maintaining (or improving) the NLL value.

**Audience:**

Yes

**Broader Impact Concerns:**

No concerns.

**Claims And Evidence:**

Yes

**Requested Changes:**

Clarity
- I recommend that the authors add more equations instead of inline math and plain text explanations. They should also add the equation numbers, provide the relationship, and provide a diagram of the proposed architecture.
- I recommend the authors weaken their claims about large-scale properties as they have not been investigated. For example, the introduction mainly discussed the LLMs, and I expected this paper to have some experiments on this scale. Note that I still like the academic computing scale experiments to show the potential of this method, but it is overclaimed to strongly appeal this method for LLMs without scaling experiments.
- Section 1, 2nd paragraph: I'm confused about this paragraph because the paper discusses the relationship with the other related studies without explaining the proposed method. I recommend that the authors first briefly explain their method before discussing the related studies. Also, the authors may create an independent (sub)section describing the related studies.
- The contribution part in Section 1: Please add the citations about wiki-8, YoutubeMix, and MAESTRO here.
The number of Figures is not aligned with the orders appearing in the paper, and it is difficult to check the figures and corresponding documents. The figure numbers and places should be re-ordered.
- Most references are based on the pre-print (arXiv) information. Please replace it with the publication information.

**Strengths And Weaknesses:**

Strengths
- LLMs' computing cost becomes a social problem (e.g., carbon footprint)
- The method has a novel architecture inspired by the wavelet operation.
- The method shows good convergence properties in the academic computing scale experiments.

Weaknesses
- The paper has difficulty explaining their architecture. The paper provides reasonable explanations for the Wavelet introduction used in this paper (e.g., Section 3.2). However, its application to the proposed method is not easy to understand. For example, there are a lot of inline maths and plain text explanations. The equations are very high-level and not detailed. There is no equation number, and the way the equations are used in the paper is unclear. Also, it requires a diagram of the architecture and how it is used. The paper only provides a general wavelet diagram.
- The paper does not have any downstream evaluation, and it is difficult to validate the performance of the proposed method with only the loss or NLL values.
- Although this effectiveness would be valid on this scale, I'm not sure these performance benefits would be maintained with large-scale training (e.g., more capacity and more training data in recent LLMs). Thus, I feel that some of the points are over-claimed.

---

> ### Author Response · Authors · 2024-08-27
> **Reviewer t874 Response Part 1**
>
> We thank the reviewer for providing and suggesting several edits and modifications to make the paper stronger and much better as compared to the previous version. We also thank the reviewer for inputs.
>
> Summary Of Contributions:
> This paper proposes imposing a wavelet structure for a transformer-based auto-regressive model, a widely used framework in LLMs. The method was applied to text modeling, raw waveform modeling, and music MIDI modeling. The paper focuses on the academic computing scale and shows the performance improvement from the vanilla transformer in terms of the convergence speed while maintaining (or improving) the NLL value.
>
> Strengths And Weaknesses:
> Strengths
> LLMs' computing cost becomes a social problem (e.g., carbon footprint)
> The method has a novel architecture inspired by the wavelet operation.
> The method shows good convergence properties in the academic computing scale experiments.
>
> Weaknesses
> The paper has difficulty explaining their architecture. The paper provides reasonable explanations for the Wavelet introduction used in this paper (e.g., Section 3.2). However, its application to the proposed method is not easy to understand. For example, there are a lot of inline maths and plain text explanations. The equations are very high-level and not detailed. There is no equation number, and the way the equations are used in the paper is unclear. Also, it requires a diagram of the architecture and how it is used. The paper only provides a general wavelet diagram.
>
> Author Reply == We have added an algorithm now. We have added the equation number and explained in the section. We have also added a diagram to explain the connection of wavelets and our proposed method.
>
> The paper does not have any downstream evaluation, and it is difficult to validate the performance of the proposed method with only the loss or NLL values.
>
> Author  Reply == We have benchmarked our method against popular Long-Range Arena benchmarks. We retained the setup exactly the same, took a lightweight causal decoder only architecture and reported the results with/without adding wavelet inspired structure to it. It gave a significant non-trivial performance boost as compared to our baseline as well as the delta increase is large as compared to other numbers reported. Further on ListOps which wants a hierarchical structure, this method shines.
>
> Although this effectiveness would be valid on this scale, I'm not sure these performance benefits would be maintained with large-scale training (e.g., more capacity and more training data in recent LLMs). Thus, I feel that some of the points are over-claimed.
>
> Author  Reply == We agree that the effectiveness is valid on this scale. One can argue one way or the other if the performance benefits would be maintained with large-scale training. Coming up with these numbers itself required significant computation for the paper. We stand-by all of the results reported in the paper. We have added extra experiments for depth and embedding dimension. Since scaling laws have been observed for continued increase in model capacity, one could argue in the favor of the paper that by enforcing such a hierarchical structure which is present in most of the data around us, we are forcing the model to learn embeddings in these slots -- like what would be optimal way to combine two tokens, four tokens, eights tokens best suited for next token prediction. Then by allowing these coarse/fine kernels to be learnable, we allow more flexibility on the latent space coordinates in guiding them for the appropriate structure. The network is forced to learn the embeddings in this manner now, which are representative of the input at different scales, and it comes with the best possible solution !  We are of the opinion that it should boost >1B parameter LLMs as well. The other side of this is -- as model size increases these behaviors emerge. As of now, these are beyond our scope and we can only speculate.

---

> ### Author Response · Authors · 2024-08-27
> **Reviewer t874 Response Part 2**
>
> I recommend that the authors add more equations instead of inline math and plain text explanations. They should also add the equation numbers, provide the relationship, and provide a diagram of the proposed architecture.
>
> Author  Reply == We have added  equations, and explanations, with equation numbers and relationship with that of our method and proposed architecture is also given.
>
> I recommend the authors weaken their claims about large-scale properties as they have not been investigated. For example, the introduction mainly discussed the LLMs, and I expected this paper to have some experiments on this scale. Note that I still like the academic computing scale experiments to show the potential of this method, but it is overclaimed to strongly appeal to this method for LLMs without scaling experiments.
>
> Author  Reply == It has been toned down in several places. As mentioned before, we can only speculate as both of us do not know what will happen if we increase the model architecture to say a 1 billion parameter model. However with the evidence at hand, it is the best one could do in an academic setup. The direction is important as well, as it is trying to push for clever techniques retaining the context length and parameters and improving the internal circuitry.
>
> Section 1, 2nd paragraph: I'm confused about this paragraph because the paper discusses the relationship with the other related studies without explaining the proposed method. I recommend that the authors first briefly explain their method before discussing the related studies. Also, the authors may create an independent (sub)section describing the related studies.
>
> Author  Reply == We have not created an independent subsection but just added a brief intro to our proposed method before adding and discussing related studies so that the reader is aware of what is going to come.
>
> The contribution part in Section 1: Please add the citations about wiki-8, YoutubeMix, and MAESTRO here. The number of Figures is not aligned with the orders appearing in the paper, and it is difficult to check the figures and corresponding documents. The figure numbers and places should be re-ordered.
>
> Author  Reply ==  We have added the references/citations in Section 1 as suggested. We have redone the Figures and changed the current ordering and added brief explanations to the caption and the paper. We have also resolved the figure numbers.
>
> Most references are based on the pre-print (arXiv) information. Please replace it with the publication information.
>
> Author  Reply == The paper has been updated to have publication information in most of the places. In other places where there was no published version/ could not find the publication details, the arxiv version is retained.
>
> Broader Impact Concerns:
> No concerns.
> Claims And Evidence: Yes
> Audience: Yes

---

### Decision · Action_Editor_5krx · 2024-09-13

**Recommendation:** Reject

**Comment:**

I recommend rewriting this paper to focus on the use of temporal smoothing to improve Transformer decoder models, since this would help with the clarity of the writing (no need to discuss wavelets at all), and would set readers' expectations more appropriately.

One of the reviewers pointed out that the LRA leaderboard shows that the Mega model (https://arxiv.org/pdf/2209.10655), which is also a Transformer with enhancements similar to those proposed in this paper, does even better on ListOps, Text, and Image, so that paper really should also be discussed as related work.

The comparison suggested by Reviewer ZDii to compare to an exponential moving average makes sense to me, since an EMA filter is simply a first-order IIR lowpass filter, the Haar wavelets used in this work are FIR lowpass filters, and it would be possible to use a range of EMA filter coefficients just like a range of Haar filter lengths are used.

**Audience:**

The core experimental results of this paper are quite interesting, and would be of interest to at least part of TMLR's audience.

**Claims And Evidence:**

This paper proposes a change to the Transformer decoder in which half of the dimensions in the output embedding are explicitly smoothed using fixed-length moving average filters applied causally. Small-scale experiments (runnable on resources available in an academic environment) show that this modification accelerates convergence of the model, measured in terms of validation loss for GPT-2 style models trained on text, symbolic music, and raw audio prediction tasks. Additionally, experiments on three long range arena tasks demonstrate that this modification improves over a standard Transformer baseline and on two tasks gives better performance than previous Transformer baselines.

If the paper framed its claimed contributions the way I have described above, it would be stronger and likely would have enjoyed a better reception from the reviewers. However, the paper frames its contribution as bringing ideas from wavelet processing into large language models (LLMs), which causes a few problems.
1. It sets the reader's expectation that the experiments will follow the standard pre-train, fine-tune, and evaluate on downstream tasks pattern seen in many LLM papers. However, due to limitations on computational resources, the authors quite reasonably focus only on the initial pre-training stage. This led to two reviewers complaining about the absence of fine-tuning results / results on downstream tasks.
2. Similarly, discussing LLMs leads readers to expect that text modeling will be done using units such as BPE or SentencePiece, while this paper uses characters for simplicity.
3. The notion that signals like text, music, and audio will have a range of temporal dependencies that might be exploited by working with temporally smoothed representations is pretty intuitive, and does not require all of the mathematical machinery of the discrete wavelet transform. The extended discussion of wavelets is thus, in my opinion, a distraction from the main point of the paper.
4. In the discussion with the reviewers, the authors justify the use of the wavelet framework by pointing to the experiments in which the parameters of the kernels are learnable (§4.4); however, this argument is not convincing because there's no guarantee that the learned kernels are even low-pass filters, much less filters appropriate for use in a discrete wavelet transform.

So, my conclusion is that the claims of the paper are not fully supported.

**Resubmission Of Major Revision:**

The authors may consider submitting a major revision at a later time.